# Comprehensive Metabolite Profiling of Berdav Propolis Using LC-MS/MS: Determination of Antioxidant, Anticholinergic, Antiglaucoma, and Antidiabetic Effects

**DOI:** 10.3390/molecules28041739

**Published:** 2023-02-11

**Authors:** Hasan Karagecili, Mustafa Abdullah Yılmaz, Adem Ertürk, Hatice Kiziltas, Leyla Güven, Saleh H. Alwasel, İlhami Gulcin

**Affiliations:** 1Department of Nursing, Faculty of Health Sciences, Siirt University, Siirt 56100, Turkey; 2Faculty of Pharmacy, Department of Pharmaceutical Chemistry, Dicle University, Diyarbakır 21280, Turkey; 3Department of Pharmacy Services, Hınıs Vocational School, Ataturk University, Erzurum 25600, Turkey; 4Department of Chemistry, Faculty of Science, Ataturk University, Erzurum 25240, Turkey; 5Department of Pharmacy Services, Vocational School of Health Services, Van Yuzuncu Yil University, Van 65080, Turkey; 6Department of Pharmaceutical Botany, Faculty of Pharmacy, Ataturk University, Erzurum 25240, Turkey; 7Department of Zoology, College of Science, King Saud University, Riyadh 11362, Saudi Arabia

**Keywords:** propolis, antioxidant activity, α-glycosidase, acetylcholinesterase, carbonic anhydrase, phenolic compound

## Abstract

Propolis is a complex natural compound that honeybees obtain from plants and contributes to hive safety. It is rich in phenolic and flavonoid compounds, which contain antioxidant, antimicrobial, and anticancer properties. In this study, the chemical composition and antioxidant activities of propolis were investigated; ABTS^•+^, DPPH^•^ and DMPD^•+^ were prepared using radical scavenging antioxidant methods. The phenolic and flavonoid contents of propolis were 53 mg of gallic acid equivalent (GAE)/g and 170.164 mg of quercetin equivalent (QE)/g, respectively. The ferric ion (Fe^3+^) reduction, CUPRAC and FRAP reduction capacities were also studied. The antioxidant and reducing capacities of propolis were compared with those of butylated hydroxyanisole (BHA), butylated hydroxytoluene (BHT), α-tocopherol and Trolox reference standards. The half maximal inhibition concentration (IC_50_) values of propolis for ABTS^•+^, DPPH^•^ and DMPD^•+^ scavenging activities were found to be 8.15, 20.55 and 86.64 μg/mL, respectively. Propolis extract demonstrated IC_50_ values of 3.7, 3.4 and 19.6 μg/mL against α-glycosidase, acetylcholinesterase (AChE) and carbonic anhydrase II (hCA II) enzyme, respectively. These enzymes’ inhibition was associated with diabetes, Alzheimer’s disease (AD) and glaucoma. The reducing power, antioxidant activity and enzyme inhibition capacity of propolis extract were comparable to those demonstrated by the standards. Twenty-eight phenolic compounds, including acacetin, caffeic acid, p-coumaric acid, naringenin, chrysin, quinic acid, quercetin, and ferulic acid, were determined by LC-MS/MS to be major organic compounds in propolis. The polyphenolic antioxidant-rich content of the ethanol extract of propolis appears to be a natural product that can be used in the treatment of diabetes, AD, glaucoma, epilepsy, and cancerous diseases.

## 1. Introduction

Propolis is a naturally occurring bee resin used by honeybees as a sealant for hexagonal cells as well as for protection from external threats and climatic conditions. Its biological features support a more aseptic environment, which may be obtained by embalming the bodies of invading species, whereas its mechanical properties fill cracks and promote thermal isolation, which aids in hive defense [1,2]. Geographical locations, botanical sources, and bee species all have a significant impact on the chemical makeup of propolis, which is influenced by several variables, including the area’s floristic makeup, location, and collection time [3]. Owing to the existence of phenolic, terpenoid and alkaloids components, propolis gathered from honeybees exhibits bioactive, antibacterial, antiviral, anesthetic, antiseptic and antioxidant characteristics [4]. Overall, this complex combination is composed of approximately 50% resins, 30% wax, 10% essential oils, 5% pollen, and 5% other substances and materials, including organic compounds [5,6]. Propolis is a naturally occurring bee resin that has been used in folk remedies for many years. It was first mentioned in ancient Egyptian history on the Eberly papyrus, was used to clean the umbilical cord of newborns during the Middle Ages, and used as an antibiotic or antiseptic during World War II. It is now a natural product that is consumed worldwide and has historically been used for systemic and oral disorders as an anti-inflammatory, antibacterial, and antifungal treatment, among other ailments [7].

Propolis also has significant concentrations of minerals including copper, iron, calcium, zinc, cobalt, and potassium as well as vitamins such as B, E, C, and A [8]. Many foods use propolis, a natural substance of plant origin, as a natural preservative. For instance, it may successfully prevent oxidation and alter the quality criteria when applied to traditional Turkish sausage and dairy drinks [9]. Moreover, propolis is a non-toxic product. Approximately 70 mg per day (1.4 mg/kg body weight per day) would be a healthy quantity for humans [10]. Propolis has been increasingly utilized in the food and cosmetic industries, in addition to being used for home treatments [11].

Reactive oxygen species (ROS) are produced during oxidative stress and endogenous antioxidants in the body are unable to limit their formation [12,13]. An imbalance between endogenous antioxidant systems and ROS increases the risk of oxidative stress in the body, which can cause several diseases. ROS may serve a triggering function in the pathogenesis of inflammation, even though the inflammatory route is complex. Therefore, preventing ROS generation may be a crucial step in reducing inflammation [14]. Owing to its radical scavenging action, propolis has become a viable source of natural products in recent years [15,16]. Oxidative deterioration of lipids, proteins, and nucleic acids in living cells is enhanced by ROS. It is generally recognized that ROS, often known as free radicals, are major contributors to the onset of many serious medical conditions, including cancer, heart disease, and aging [17,18,19]. In living organisms, antioxidants delay thwart, and suppress the oxidation of lipids, carbohydrates, nucleic acids, and proteins. They contain phenols and polyphenols, which are powerful substances that reduce or eliminate the harmful effects of ROS [20]. Furthermore, antioxidants have frequently been employed as dietary supplements to protect foods from oxidative deterioration [21]. Antioxidants can halt the progression of many chronic illnesses and lipid peroxidation, shielding the human body from the damaging effects of free radicals and ROS. As a result, there is much interest in natural additives as substitutes for the most widely used antioxidants, including propyl gallate, BHA, BHT, and *tert*-butylhydroquinone. Liver damage and cancer development have been linked to BHA and BHT [22]. As a result, there is increasing interest in natural antioxidants and their use in culinary applications is growing [23,24].

The α-glycosidase enzyme, whose activity is critical for the breakdown of dietary polysaccharides, is one of the key targets in the treatment of diabetes [25]. α-Glycosidase inhibitors (AGIs) prevent the absorption of monomeric sugar units in the intestine; this reduces the level of postprandial plasma glucose levels. Therefore, such inhibitors can be used to treat diabetes and obesity [26,27]. The age-related neurodegenerative AD condition, which is sometimes referred to as type 3 diabetes owing to its pathophysiological resemblance to type 2 diabetes mellitus (T2DM), is triggered by both inflammatory diseases and T2DM [28].

AD is a neurological condition that progresses abruptly and causes behavioral changes, forgetfulness, cognitive impairment, and deficiencies in language. Acetylcholine (ACh) is broken down by acetylcholinesterase (AChE) into acetate (CH_3_COO^–^) and choline (Ch) [29,30]. Furthermore, the use of AChE inhibitors (AChEIs) to prevent the cholinergic breakdown of ACh is a prospective strategy for treating AD [31]. Newer, efficient and safe AChE research is required to address neurological damage since AChEIs have substantial adverse effects. Anticholinesterases or AChEIs are cholinesterase enzyme inhibitors [32]. AChEIs are often used in medicine, particularly for the treatment of AD. AChEIs and prospective lead molecules for AD have both been identified as phenolic chemicals [33,34].

Carbonic anhydrases (CAs) are metalloenzymes that contain Zn^2+^ and catalyze the reversible hydration of carbon dioxide (CO_2_) to protons and bicarbonate (HCO_3_^−^) [35,36]. CAs play a variety of biochemical and metabolic roles, including ureagenesis, lipogenesis, and gluconeogenesis [37,38,39]. They also maintain a fluid balance throughout the body, particularly in the eyes, stomach, and kidneys. Glaucoma-related elevated intraocular pressure (IOP) can be relieved or treated with carbonic anhydrase inhibitors (CAIs) [40]. Various analytical techniques have been used to determine the quality of propolis. The commonly used techniques include UV spectrophotometry and LC-MS/MS.

The present study aimed to investigate the chemical ingredients and biological activities of propolis collected from the East Anatolian region of Turkey. To achieve this, the following was performed: (a) flavonoid and phenolic profiles of the samples were evaluated using LC-MS/MS; (b) the total antioxidant/phenolic capabilities of propolis equivalents were measured using DPPH, ABTS, DMPD, CUPRAC, ferric-reducing antioxidant power (FRAP), and Folin–Ciocalteu techniques; (c) the inhibitory effect of propolis on some metabolic enzymes including AChE, hCA II and α-glycosidase were investigated to determine a probable relationship with AD, diabetes mellitus, and glaucoma.

## 2. Results

### 2.1. Phenolic Contents of Propolis

The total phenolic and flavonoid contents were evaluated for propolis. In this context, 53.0 mg of gallic acid equivalent (GAE)/g and 170.2 quercetin equivalent (QE)/g were calculated in propolis extract. Using 53 phenolic compounds as standards, the LC-MS/MS method was utilized to identify the major organic components in propolis preparations (Figure 1). Phenolic compounds were elucidated by comparing their chromatographic behavior, UV spectra, and MS information with reference compounds, and 28 compounds were discovered (Table 1). Table 2 lists the mean values of each chemical based on LC-MS/MS analysis.

The major components detected in propolis were the flavone acacetin (76.359 mg/g), caffeic acid (21.358 mg/g), p-coumaric acid (16.911 mg/g), naringenin (11.34 mg/g), chrysin (9.86 mg/g), quinic acid (7.285 mg/g), quercetin (6.223 mg/g), ferulic acid (5.11 mg/g), apigenin (4.686 mg/g), luteolin (4.394 mg/g), kaempferol (4.043 mg/g), hesperidin (2.089 mg/g), vanillic acid (1.647 mg/g) and protocatechuic acid (1.158 mg/g) (Table 2).

### 2.2. Reducing Abilities Results

The ferric ion (Fe^3+^) reductive abilities of the isolated phenolic compounds from propolis were determined according to the Oyaizu method [41]. The addition of Fe^3+^ to the compound causes Fe_4_[Fe(CN^–^)_6_]_3_ complex formation, which results in maximum absorption at 700 nm [42,43,44]. As summarized in Table 3, the propolis extract exhibited a potent Fe^3+^ reducing profile. However, the Fe^3+^-reducing ability of a concentration of 30 μg/mL of propolis and standards decreased in the following order: BHT (2.018, r^2^: 0.9466) > α-tocopherol (1.895, r^2^: 0.9402) > Trolox (1.545, r^2^: 0.9966) > BHA (1.257, r^2^: 0.9523) > Propolis (0.894, r^2^: 0.9953). All analyses were performed in triplicate.

The cupric ion (Cu^2+^)-reducing abilities of the phenolic composition in propolis are presented in Table 3. There was a strong relationship between the Cu^2+^-reducing effect and different concentrations of propolis. However, at 30 µg/mL, significant absorbance of reducing power was demonstrated by the propolis. In contrast, the Cu^2+^-reducing ability of propolis and the standards were found to be BHT (2.912, r^2^: 0.9969) > Trolox (2.323, r^2^: 0.9980) > BHA (1.800, r^2^: 0.9742) > α-Tocopherol (1.139, r^2^: 0.9967) > Propolis (0.778, r^2^: 0.9986). The CUPRAC antioxidant method is convenient, selective, inexpensive, fast, and stable [45]. According to the results provided in Table 3, the following was found: BHT (2.089, r^2^: 0.9581) > α-Tocopherol (1.995, r^2^: 0.9807) > Trolox (1.755, r^2^: 0.9990) > Propolis (1.114, r^2^: 0.9970) > BHA (0.884, r^2^: 0.9899) (Table 3). The higher the absorbance readings in this approach, the greater the reducing ability of the test samples.

### 2.3. Radical Scavenging Results

The DPPH^•^ scavenging activity of propolis was measured, and the half maximal inhibition concentration (IC_50_) value was determined (Table 3). Propolis demonstrated concentration-dependent radical scavenging activity. Propolis revealed comparable and stronger anti-radical activities (20.55 µg/mL) than BHT (21.0 μg/mL), but had a better antioxidant potential in propolis samples taken from 39 different locations in Turkey. These propolis samples had antioxidant capabilities ranging from 55.98 ± 0.02% to 86.17 ± 0.16% [46]. In addition, our sample has a higher antioxidant capacity compared to various European propolis specimens (26.45 μg/mL; 27.72 μg/mL) [47]. The IC_50_ values of Chinese propolis samples ranged greatly, from 71.19 ± 5.31 µg/mL to 432.08 ± 6.42 µg/mL, showing that the antioxidant activity of these propolis is not only lower but also region-dependent [48]. All analyses were performed in triplicate. Propolis IC_50_ values and reference radical scavenger agents such as Trolox, α-tocopherol, BHT, and BHA were 8.157 μg/mL for propolis, 7.71 μg/mL for BHT, 7.71 μg/mL for BHA, 7.71 μg/mL for Trolox, and 8.10 μg/mL for α-tocopherol (Table 3). As indicated in Table 4, propolis was efficient in DMPD^•+^ scavenging at concentrations ranging from 10 to 30 μg/mL. The IC_50_ of propolis was 86.64 μg/mL. The value obtained for this was 31.43 μg/mL for BHA, 14.38 μg/mL for Trolox. At all propolis concentrations, the concentration of DMPD^•+^ decreased significantly (*p* < 0.01).

### 2.4. Enzyme Inhibition Results

The hCA II isoform is associated with some disorders, including glaucoma, osteoporosis, and renal tubular acidosis. CA inhibitory effects of the propolis were tested and the results are presented in Table 5 and Figure 2A. IC_50_ values were 19.6 μg/mL (r^2^: 0.9327) for propolis towards CA II. For acetazolamide, the IC_50_ was 8.37 μg/mL (r^2^: 0.9825), which was used as a control for the carbonic anhydrase isoenzyme inhibition experiment [49].

The inhibition level of propolis extract was comparable to that of tacrine, a common reference inhibitor of the AChE. Table 5 summarizes the IC_50_ values of the propolis extract for enzyme inhibition. The IC_50_ value (3.4 µg/mL) of the ethanol extract for propolis against AChE showed a higher inhibition effect compared to that of tacrine (Table 5). The IC_50_ of tacrine was 5.97 µg/mL (r^2^: 0.9706), which was utilized as a control for the AChE inhibition experiment (Figure 2B).

Propolis extracts displayed an IC_50_ value of 3.7 μg/mL towards α-glycosidase (r^2^: 0.9362, Table 5 and Figure 2C). The results reveal that propolis has inhibitory effects similar to those of α-glycosidase efficient acarbose (IC_50_: 22800 nM) as a typical glycosidase inhibitor [50]. In the present study, the α-glycosidase enzyme inhibition effect of propolis was higher than that of propolis samples taken from various locations in Morocco (IC_50_: 90.99–876.24 μg/mL) of [51].

## 3. Discussion

Phenolic chemicals present in all plants are vital components of the human diet. They have received a great deal of attention because of their biological activities, which include antioxidant characteristics [52,53,54]. Propolis contains an appreciably high number of phenolic compounds, which have positive effects on human health. Phenolic compounds also hinder oxidation and improve the chemical stability of food products [55]. Flavonoids, such as flavones, flavonols, flavanones, and dihydroflavonols, as well as other polyphenols, are primarily responsible for the biological action of propolis [16,56,57]. Caffeic acid phenylethyl ester, gallic acid, cinnamic acid, galangin, caffeic acid, naringenin, luteolin, kaempferol, quercetin, pinocembrin, rutin, p-coumaric acid, and ferulic acid are among the components of propolis. They ultimately enhance the effective digestion, antioxidant capacity, and metabolic, physiological, and immune capabilities of body tissues [58,59]. More than 300 substances, including phenolic acids and their esters, flavonoids, terpenes, triterpenes, alcohols, aromatic aldehydes, fatty acids, stilbenes and steroids, lignans, amino acids, and sugars, among others, have been observed in various propolis species [60]. Considering the phenolic content determinations of propolis performed as gallic acid equivalent, the propolis samples obtained from different locations in Turkey displayed effective phenolic (88.7–261.1 mg GAE/g propolis) and total flavonoid (37.5–150.4 mg QE/g propolis) contents of the ethanolic propolis extracts [61]. In another study, aqueous extract of propolis expressed 124.3 μg GAE and 8.15 μg QE per g of aqueous extract of propolis [2]. In addition, Aygul and co-workers determined that Ankara propolis (8.50 mg GAE/g propolis) and Giresun propolis (7.88 mg GAE/g propolis) exhibited notable total phenolic contents [62].

Therefore, it is important to choose the most appropriate method to determine the antioxidant capacity of herbal extracts or biological samples. In the present study, many bioanalytical methods, such as reducing effects and radical removal methods, were used to determine the antioxidant capacity of the extract [63,64]. The diversity, high number of ingredients, and rich phenolic content may explain the antioxidant potential of propolis. The reduction potentials of phenolic compounds isolated from propolis were determined using three different reduction systems, including Fe^3+^, CUPRAC, and Fe^3+^-TPTZ ion-reducing abilities [65,66]. The radical scavenging properties of the propolis were examined using DPPH, ABTS, and DMPD radical scavenging assays. Plants, natural compounds, and propolis samples can exhibit reducing properties, thereby neutralizing oxidants and ROS.

The total reducing capacity of the pure antioxidant substances and plant extracts was determined using the FRAP test. Ferric salt, which is the basis of the FRAP test, was utilized as an oxidant in the electron transfer process [67]. Owing to its colored combination with TPTZ, which has a maximum absorbance at 593 nm, Fe^2+^ may be detected spectrophotometrically [68]. Depending on the reducing power of the antioxidant samples, the yellow color of the test solution changed to different colors of green or blue in this assay. The reducing capacity of a compound may be a good predictor of its potential antioxidant activity. The Fe^3+^-TPTZ-reducing test was used to assess the reducing capabilities of propolis and standards [69]. In our previous study, we determined that the aqueous extract of propolis exhibited concentration-dependent (10–30 μg/mL) Fe^3+^-reducing and cupric ion (Cu^2+^)-reducing abilities with statistically significant differences (*p* < 0.01) [2].

The DPPH method is based on the DPPH^•^ removal of antioxidant components in the plant extracts. The scavenging effect of ABTS^•+^ radicals is based on a similar mechanism [70,71]. The DPPH^•^ test, which is based on reducing DPPH^•^ to the non-radical form DPPH-H, is commonly used to assess the antioxidant capacity [72,73]. Propolis is thought to have natural antioxidant potential if it possesses DPPH^•^ scavenging capability. In an ABTS/K_2_S_2_O_8_ system, radicals of ABTS were produced [74]. This test is a decolorization approach in which the radical is created directly in a stable state prior to treatment with suspected antioxidants. The improved approach for producing ABTS^•+^ reported here involves the direct creation of a blue–green ABTS^•+^ chromophore via the reaction between ABTS and potassium persulfate [75]. A prior study revealed that aqueous extract of propolis displayed effective DPPH^•^ radical scavenging with an IC_50_ value of 31.81 μg/mL [2]. 

When compared to positive controls, the data clearly revealed that propolis has effective ABTS^•+^ scavenging activity. Our sample showed a higher ABTS^•+^ scavenging effect than that of lyophilized aqueous extract of propolis from Turkey’s Erzurum province (IC_50_: 14.29 μg/mL) [2]. A lower IC_50_ value similar to the DPPH free radical scavenging activity suggests higher ABTS^•+^ scavenging activity. According to previous reports, the principal disadvantage of the DMPD^•+^ approach is that its sensitivity and repeatability are significantly reduced when hydrophobic antioxidants such as α-Tocopherol or BHT are utilized [76]. Considering the literature, it seems that Moroccan propolis, which was collected from different locations, exhibited effective superoxide anion and nitric oxide radicals scavenging activity and metal chelating properties [77]. Our previous study demonstrated that aqueous extract of propolis had effective DMPD^•+^ (IC_50_: 18.32 μg/mL) and superoxide anion (O_2_^•-^, IC_50_: 9.89 μg/mL) radicals scavenging activities [2].

α-Glycosidase suppression causes delays in sugar absorption during digestion. Clinical trials using acarbose and miglitol as α-glycosidase inhibitors have revealed lower postprandial hyperglycemia and greater insulin sensitivity [78]. These inhibitors block the α-glycosidase enzyme in the small intestine, which is responsible for the digestion of complex carbohydrates. This enzymatic process lowers postpartum blood glucose levels by reducing carbohydrate breakdown and glucose absorption [79]. When the literature was searched, it was observed that Moroccan propolis collected from different regions inhibited the α-glycosidase enzyme with IC_50_ values between 0.01–0.07 mg/mL. Moreover, it was observed that the same propolis samples inhibited α-amylase as another digestive enzyme, with IC_50_ values between 0.09 and 0.52 mg/mL [77].

AD is the most prevalent neurodegenerative ailment, and the leading cause of dementia among the elderly. The reduction in AChE levels in the brain is the most significant biochemical alteration in AD [80]. AChEIs are used for the treatment of AD; however, these drugs have several negative side effects. As a result, research and use of novel potent antioxidants and AChE agents are greatly needed [81]. It was also found that the predominant AChE inhibitory effects were related to aromatic chemicals, and to a lesser extent, aliphatic molecules [82]. Medicinal herbs are always rich in cholinesterase inhibitors [83]. The cholinesterase inhibitory properties of propolis extract were evaluated in the current study using AChE. The ethanol extract was shown to effectively inhibit AChE. In a previous study, the inhibition effects of propolis, which was obtained from different locations, on some crucial enzymes, such as urease, xanthine oxidase and AChE, were investigated. They found that the propolis sample inhibited AChE enzyme with IC_50_ values ranging from 0.221 to 1340 mg/mL [61]. Previous studies have shown that there was a relationship between propolis phenolic contents and AChE inhibition, which importantly suggests that the enzyme was probably inhibited by phenolic substances [61]. It is known that propolis is a complex resinous material. Therefore, it is very important that the specific phenolics in this complex mixture exhibit high AChE inhibitions even at low concentrations.

CA II has been linked to epilepsy, glaucoma, edema, and assumable altitude sickness [84]. The activation and inhibition of CA isoforms have important therapeutic goals in the treatment of a variety of disorders including glaucoma, edema, cancer, obesity, hypertension, epilepsy, and osteoporosis [85]. CA II suppression reduces HCO_3_^−^ generation and, as a result, aqueous humor secretion, resulting in lower ocular pressure [86]. Glaucoma is a multifactorial optical disease characterized by optical nerve degeneration, which is mostly associated with high IOP, which can result in blindness. Because hCA inhibitors such as acetazolamide, brinzolamide, and dorzolamide are effective in lowering IOP after topical treatment, novel therapeutic considerations are required [87]. In another study on propolis, it was determined that Ankara propolis (IC_50_: 1.273 μg/mL) and Giresun propolis (IC_50_: 1.374 μg/mL) had quite high cytosolic hCA I isoform. On the other hand, both propolis samples inhibited predominant and cytosolic hCA II isoenzyme with IC_50_ values of 0.486 and 0.612 μg/mL [62].

## 4. Materials and Methods

### 4.1. Chemicals

Acetylcholinesterase, acetylcholine iodide, α-glycosidase, p-nitrophenyl-D-glycopyranoside, 2,2′-azino-bis 3-ethylbenzthiazoline-6-sulfonic acid (ABTS), 1,1-diphenyl-2-picrylhydrazyl (DPPH), N,N-dimethyl-p-phenylenediamine (DMPD), 2,9-dimethyl-1,10-phenanthroline (Neocuproine), butylated hydroxytoluene (BHT), butylated hydroxyanisole (BHA), α-tocopherol, trolox, and standard phenolic compounds of LC-MS/MS were purchased from Sigma (Sigma-Aldrich GmbH, Steinheim, Germany). The other materials were procured from Sigma-Aldrich or Merck (Darmstadt, Germany), appropriately. Propolis was dissolved in ethanol for antioxidant activities, but in DMSO for enzyme inhibition tests due to the potential inhibitory effects of ethanol.

### 4.2. Preparation of Propolis

A sample of propolis (50 g) was collected in May 2022 from one of the beehives of Yuksel Gulcin, a farmer located in Berdav, a village in Tutak in the district of Agri, and stored before processing. The extraction process was completed as previously mentioned [2,88]. For preparation of propolis, a 25 g sample was milled into a fine powder and combined with ethanol. The prepared extract was filtered using Whatman No.1 paper and the filtrate was collected before removing the ethanol using a rotary evaporator (RE 100 Bibby, Stone Staffordshire, England) at 50 °C to obtain a dry extract. The yield of propolis was calculated as 75% and 18.75 g extract was placed in a dark plastic bottle and kept at −20 °C until use.

### 4.3. Determination of Total Soluble Phenolic Contents of Propolis

Total phenol content was used to determine the amount of phenolic compound present in propolis as gallic acid equivalents. [89]. The procedure was based on that described by Singleton and Rossi [90], with slight modifications [91]. Propolis extract (0.5 mL) was added to 1.0 mL of Folin–Ciocalteu reagent [92] as described in detail in [93]. Afterward, carbonate (0.5 mL, 1%) was added and the mixture was stirred vigorously. Absorbance was measured at 760 nm against a water-containing blank sample after 2 h of incubation in the dark at room temperature. The quantity of phenol in one gram of propolis extract was calculated as mg of gallic acid equivalents (GAE).

### 4.4. Determination of Tatal Flavonoid Content of Propolis

The total flavonoid content was determined using the aluminum chloride (AlCl_3_) technique [94]. Briefly, 0.5 mL of the propolis extraction solution was combined with 1.5 mL of 95% methanol, 1.5 mL of 10% AlCl_3_, 0.5 mL of 1.0 M potassium acetate solution, and 2.3 mL of distilled deionized water. The absorbance was measured at 415 nm after incubation in the dark (25 °C, 40 min). Water was used as the blank sample. The total amount of flavonoids was calculated as mg quercetin equivalents (QE)/g of propolis extract [95].

### 4.5. Test Solution for Mass Spectrometer (LC–MS/MS) and Chromatography Conditions

The analytical approach used in this investigation was conducted in accordance with recent research. The Dicle University Central Research Laboratory conducted the LC-MS/MS experiments. The analytical approach employed in this study was designed by Yılmaz [96] and adapted for propolis. To investigate the phytochemical component in propolis, the following 53 phytochemical standards were purchased from Sigma-Aldrich (Steinheim, Germany): quinic acid, fumaric acid, gallic acid, malic acid, aconitic acid, epigallocatechin, proto-catechuic acid, catechin, gentisic acid, chlorogenic acid, pro-tocatechuic aldehyde, tannic acid, epigallocatechin gallate, 4-OH benzoic acid, epicatechin, vanillic acid, caffeic acid, syringic acid, vanillin, syringic aldehyde, daidzin, epicatechin gallate, piceid, ferulic acid, p-coumaric acid, sinapic acid, coumarin, salicylic acid, cynaroside, cinnamic acid, rutin, isoquercitrin, hesperidin, o-coumaric acid, genistin, rosmarinic acid, ellagic acid, cosmosiin, quercitrin, astragalin, nicotiflorin, fisetin, cynarine, daidzein, quercetin, naringenin, luteolin, hesperetin, genistein, apigenin, kaempferol, amentoflavone and chrysin, which were used to investigate the thephytochemical component in propolis. 1,5-Dicaffeoylquinic acid and the internal standards ferulicacid-D3, rutin-D3, and quercetin-D3 were purchased from TRC (Toronto, Canada). A Shimadzu–Nexera model ultrahigh performance liquid chromatograph (UHPLC) (Shimadzu, Japan) connected to a tandem mass spectrometer was used to quantify the 53 phytochemicals. An autosampler (SIL-30AC model), column oven (CTO-10ASvp type), binary pumps (LC-30AD model), and degasser were installed on the reversed-phase UHPLC (DGU-20A3R model). The chromatographic settings were tuned to provide the optimal separation of the 53 phytochemicals while overcoming the suppression effects. The following parameters were tested and applied until the optimum conditions were achieved: different columns such as the Agilent Poroshell 120 EC-C18 model (150 mm × 2.1 mm, 2.7 µm) and RP-C18 Inertsil ODS-4 (100 mm × 2.1 mm, 2 µm), different mobile phases (B) such as acetonitrile and methanol, different mobile phase additives such as ammonium format, formic acid, ammonium acetate, and acetic acid, and different column temperatures (25, 30, 35, and 40 °C). Consequently, chromatographic separation was performed on a reversed-phase Agilent Poroshell 120 EC-C18 model analytical column. The column temperature was set to 40 °C. The elution gradient comprised eluent A (water + 5 mM ammonium formate + 0.1% formic acid) and eluent B (methanol + 5 mM ammonium formate + 0.1% formic acid). The following gradient elution profiles were used: 20–100% B (0–25 min), 100% B (25–35 min), and 20% B (35–45 min). The solvent flow rate and injection volume were settled to 0.5 mL/min and 5 µL, respectively. Mass spectrometric detection was performed using a Shimadzu LCMS-8040 model tandem mass spectrometer equipped with an electrospray ionization (ESI) source operating in both negative and positive ionization modes. LC-ESI-MS/MS data were acquired and processed using LabSolutions software (Shimadzu). Multiple reaction monitoring (MRM) mode was used for the quantification of phytochemicals. The MRM method was optimized to selectively detect and quantify phytochemical compounds based on screening of specified precursor phytochemical-to-fragment ion transitions. The collision energies (CE) were optimized to generate optimal photochemical fragmentation and maximal transmission of the desired product ions. The MS operating conditions were applied as follows: drying gas (N2) flow, 15 L/min; nebulizing gas (N2) flow, 3 L/min; DL temperature, 250 °C; heat block temperature, 400 °C; and interface temperature, 350 °C.

### 4.6. Ferric Ions (Fe^3+^) Reducing Assay

The direct reduction of Fe^3+^(CN^—^)_6_ and the absorbance resulting from the formation of the Perl’s Prussian Blue complex upon the addition of excess ferric ions (Fe^3+^) were used to the test ferric-reducing antioxidant capacity. Thus, the FRAP method was employed to assess the lowering capacity of propolis [97]. The reduction of (Fe^3+^) ferricyanide in stoichiometric excess relative to the antioxidants is the basis for this approach. In 0.75 mL of distilled water, different doses of propolis (10–30 µg/mL) were combined with 1.25 mL of 0.2 M, pH 6.6 sodium phosphate buffer and 1.25 mL of potassium ferricyanide [K_3_Fe(CN)_6_] (1%). For 30 min, the mixture was incubated at 50 °C. After 30 min, the reaction mixture was acidified with 1 mL 10% trichloroacetic acid and incubated in the dark for 30 min. Finally, 0.25 mL of FeCl_3_ (0.1%) was added to the solution, and the absorbance at 700 nm was measured. Increased absorbance of the reaction mixture implies a greater reduction capacity [98].

### 4.7. Cupric Ions (Cu^2+^) Reducing—CUPRAC Assay

To determine the Cu^2+^-reducing antioxidant capacity of propolis, the method proposed by Apak et al. [58] was used with slight modifications. Briefly, 0.25 mL CuCl_2_ solution (0.01 M), 0.25 mL ethanolic neocuproine solution (7.5 × 10^−3^ M) and 0.25 mL CH_3_COONH_4_ buffer solution (1.0 M) were added to a test tube, which was then mixed with various concentrations of propolis (10–30 µg/mL). The whole volume was then reduced to 2 mL by adding distilled water and vigorously mixing. The tubes were sealed and stored at room temperature. After 30 min, absorbance was measured at 450 nm against a reagent blank. The increased absorbance of the reaction mixture suggests an increased reduction capacity [99].

### 4.8. Fe^3+^-TPTZ Reducing—FRAP Assay 

FRAP is based on decreasing Fe^3+^-TPTZ in acidic media [100]. The increased absorbance of the blue color of the ferrous form of the complex (Fe^2+^-TPTZ) was measured spectrophotometrically at 593 nm [101]. Briefly, 2.25 mL of the newly created TPTZ solution (10 mM) in HCl (40 mM) was poured into 2.5 mL of acetate buffer (pH 3.6, 0.3 M) and FeCl_3_ solution in water (2.25 mL, 20 mM). Then, propolis (10–50 μg/mL) was dissolved in a buffer solution (5 mL) and the mixture was incubated in the dark at 37 °C for 30 min. Finally, the absorbance of each sample was measured.

### 4.9. DPPH^•^ Scavenging Activity 

The DPPH^•^ scavenging activity of propolis was assessed using the DPPH^•^ scavenging method [102]. The DPPH solution was prepared the day before measurement. The solution flask was coated with aluminum foil, stirred for 16 h and kept in the dark at 4 °C. Briefly, a 0.1 mM DPPH solution was prepared in ethanol, and 0.5 mL of this solution was added to 2 mL of propolis sample solution in ethanol at different concentrations (10–30 µg/mL). The propolis samples were vortexed and incubated at 30 °C in the dark for 30 min. Absorbance was measured at 517 nm against blank samples. A decrease in absorbance indicated DPPH free radical scavenging activity. The decrease in absorbance indicated that DPPH actively scavenges free radicals.

### 4.10. ABTS^•+^ Scavenging Activity

The ABTS^•+^ radical scavenging activity of propolis was determined using a previous method [103]. ABTS^•+^ has a blue–green color and a distinctive absorbance at 734 nm. The reaction of ABTS (2 mM) in water and potassium persulfate (2.45 mM) at room temperature, which was vortexed for 30 min in a flask coated with aluminum foil, led to the production of the ABTS^•+^ cation radical. At 734 nm, an absorbance of 0.800 ± 0.05 was obtained by diluting the ABTS^•+^ solution with phosphate buffer (0.1 M, pH 7.4). Then, 0.25 mL of the ABTS^•+^ solution was added to 1.75 mL of the sample solution in ethanol at various propolis concentrations (10–30 µg/mL). The propolis sample was vortexed and left in the dark for 30 min. After 30 min, the absorbance at 734 nm for each concentration was measured and compared with that of the blank. The decreased absorbance of the sample suggests ABTS^•+^ cation radical scavenging activity [104].

### 4.11. DMPD^•+^ Scavenging Activity

The radical scavenging capacity of propolis against DMPD was measured using the method described by Fogliano et al. [105]. This assessment is based on the ability of the extract to suppress the production of DMPD^•+^ cation radicals. Briefly, 105 mg was added to 5 mL DMPD^•+^ solution. Then, 1 mL of this solution was added to 100 mL acetate buffer (pH 5.3, 0.1 M), and agitated for 5 min in the dark. Ferric chloride (0.2 mL, 0.05 M) was added to this solution to form DMPD^•+^. Standard antioxidants or propolis at varying doses (10–30 µg/mL) were added and the total volume was adjusted with distilled water (0.5 mL). The DMPD^•+^ solution (1 mL) was immediately added to the reaction mixture, which was thoroughly mixed and incubated for 50 min in the dark. The absorbance was measured at 505 nm [106].

### 4.12. Percentage Scavenging and IC_50_ Determination

The antioxidant (DPPH^•^, ABTS^•+^, and DMPD^•+^) scavenging potentials were calculated by comparing them to the typical antioxidant substances (BHA, BHT, Trolox, and α-tocopherol). The dose-dependent antioxidant potential of propolis was investigated using various concentrations (10–30 µg/mL) of the sample and reference standards. The propolis concentrations that caused 50% enzyme inhibition (IC_50_) values were calculated from the activity (%) versus propolis plots. First, enzyme inhibition was studied at several propolis concentrations. The obtained values were plotted as % activity against propolis concentrations. The IC_50_ values were calculated from these graphs [107].

### 4.13. AChE Enzyme Inhibition Assay

The AChE inhibitory action of propolis was assessed as previously described by Ellman et al. [108]. AChE activity was measured spectrophotometrically at 412 nm using acetylthiocholine iodide as a substrate for the enzymatic reaction, and AChE activity was measured using 5,5′-dithio-bis (2-nitro-benzoic) acid.

### 4.14. α-Glycosidase Enzyme Inhibition Assay

The inhibitory effect of these substances on α-glycosidase enzyme activity were tested using a p-nitrophenyl-D-glycopyranoside (p-NPG) substrate [50]. First, 40 µL of the sample solution was mixed with 200 µL phosphate buffer (0.15 EU/mL, pH 7.4). Furthermore, after preincubation, 50 µL p-NPG in phosphate buffer (5 mM, pH 7.4) was added and incubated again at 30 °C. Absorbance was measured spectrophotometrically at 405 nm according to a previous study.

### 4.15. hCA II Isoenzyme Inhibition Assay

The method established by Verpoorte et al. [109] previously described the inhibition of both hCA isoenzymes. The dominant cytosolic CA II isoenzyme was isolated from human erythrocytes using affinity column chromatography with Sepharose-4B-Tyrosine-sulfanylamide [110]. After loading the material into the affinity chromatography column, it was equilibrated with Tris-Na_2_SO_4_/HCl (pH 8.7, 22 mM/25 mM). CA II was eluted with 0.5 M sodium acetate/NaClO_4_ (pH 5.6, 25 °C). The differences in absorbance were measured over 3 min at 348 nm using p-nitrophenylacetate (PNA) as a substrate, which was transformed into the p-nitrophenolate ion by both isoenzymes. One enzyme unit of CA esterase activity was defined as the hydrolysis of 1 mol PNA in 1 min to *p*-nitrophenolate and acetate. The Bradford assay was used to quantify the amount of protein throughout the purification process. As a reference protein, bovine serum albumin was employed. SDS-PAGE was used to control the purity of the CA II isoform [111].

### 4.16. Statistical Analysis

All measurements were performed in triplicate for each sample. Data are presented as means (n = 3) and evaluated using one-way ANOVA followed by Tukey’s post hoc test; *p* < 0.001 was considered statistically significant.

## 5. Conclusions

Propolis, which is vital for bees themselves, their offspring and their hives, has a rich natural content obtained from different parts of the surrounding plants. This product, which it believed to be rich, nutritious and contributes to the formation of honey, has been used by mankind for thousands of years. According to the LC-MS/MS analysis, the major components detected in propolis are acacetin, caffeic acid, p-coumaric acid, naringenin, chrysin, quinic acid, quercetin, ferulic acid, apigenin, luteolin, kaempferol, hesperetin, vanillic acid, and protocatechuic acid. Furthermore, the propolis ethanol extract had increased antioxidant activity, reducing power, and phenolic contents, and inhibited AChE, α-glycosidase, and hCA II. Propolis can be used as a natural product in the treatment of serious and common T2DM, AD and glaucoma diseases, neurodegenerative, hormonal, and metabolic diseases, as well as in the food and pharmaceutical industries, owing to its phenolic and flavonoid contents, which have antioxidant, reducing and radical scavenging capacities.

## Figures and Tables

**Figure 1 molecules-28-01739-f001:**
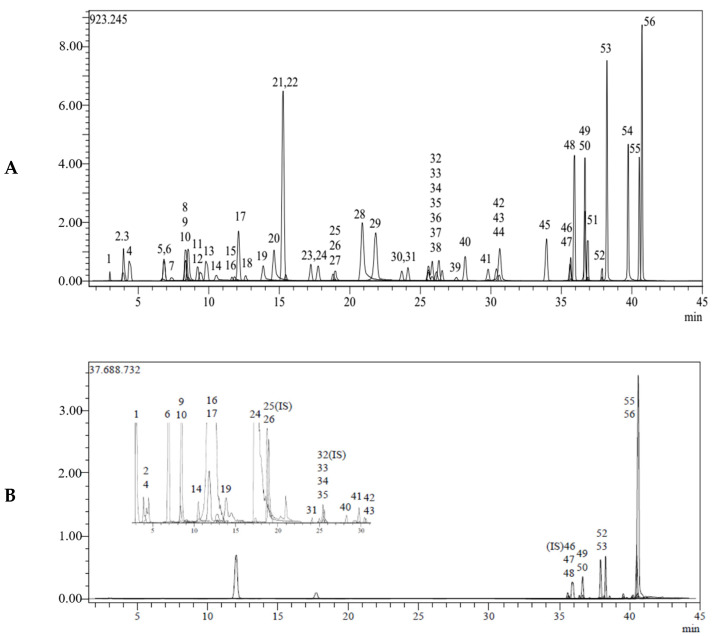
(**A**). Chromatogram of all standard phenolic compounds analyzed using the LC-MS/MS method 1: Quinic acid, 2: Fumaric acid, 4: Gallic acid 6: Protocatechuic acid, 9: Chlorogenic acid, 10: Protocatechuic aldehyde, 14: 4-OH Benzoic acid, 16: Vanillic acid, 17: Caffeic acid, 19: Vanillin, 24: p-Coumaric acid, 26: Ferulic acid, 31: Miquelianin, 33: Rutin, 34: isoquercitrin, 35: Hesperidin, 40: Cosmosiin, 41: Quercitrin, 42: Astragalin, 43: Nicotiflorin, 47: Quercetin, 48: Naringenin, 49: Hesperetin, 50: Luteolin, 52: Kaempferol, 53: Apigenin, 55: Chrysin, and 56: Acacetin. (**B**)**.** Chromatogram of propolis ethanol extract compounds.

**Figure 2 molecules-28-01739-f002:**
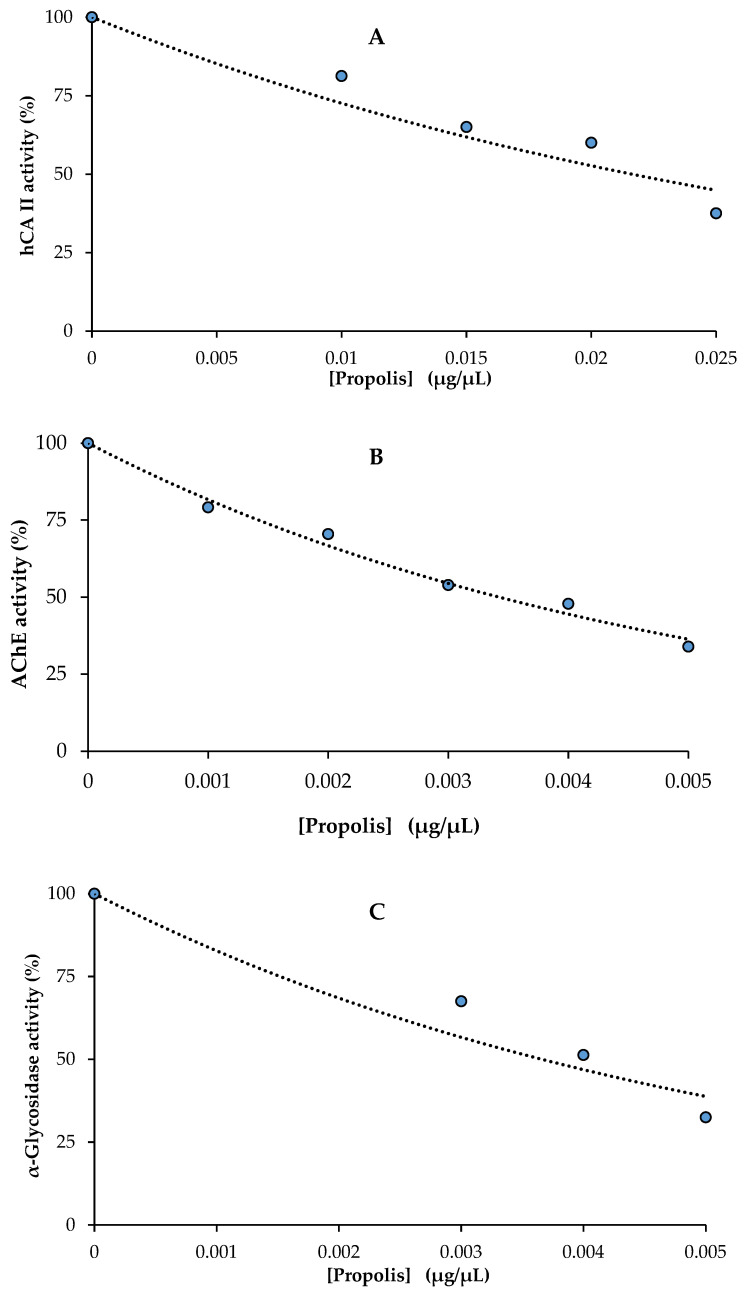
The half-maximal inhibitory concentration (IC_50_) graphs of propolis against human carbonic anhydrase II (hCA II) (**A**), acetylcholinesterase (AChE) (**B**) and α-glycosidase (**C**) enzymes.

**Table 1 molecules-28-01739-t001:** Analytical method validation parameters that belong to the LC-MS/MS method.

No.	Analytes	RT *^a^*	M.I. (m/z) *^b^*	F.I. (m/z) *^c^*	Ion. Mode	Equation	R^2 *d*^	RSD% *^e^*		Linearity	LOD/LOQ (µg/L) *^f^*	Recovery (%)		U *^g^*	Gr. No. *^i^*
								Interday	Intraday			Interday	Intraday		
1	Quinic acid	3.0	190.8	93.0	Negative	*y* = −0.0129989 + 2.97989*x*	0.996	0.69	0.51	0.1–5	25.7/33.3	1.0011	1.0083	0.0372	1
2	Fumaric aid	3.9	115.2	40.9	Negative	*y* = −0.0817862 + 1.03467*x*	0.995	1.05	1.02	1–50	135.7/167.9	0.9963	1.0016	0.0091	1
3	Aconitic acid	4.0	172.8	129.0	Negative	*y* = −0.7014530 + 32.9994*x*	0.971	2.07	0.93	0.1–5	16.4/31.4	0.9968	1.0068	0.0247	1
4	Gallic acid	4.4	168.8	79.0	Negative	*y* = 0.0547697 + 20.8152*x*	0.999	1.60	0.81	0.1–5	13.2/17.0	1.0010	0.9947	0.0112	1
5	Epigallocatechin	6.7	304.8	219.0	Negative	*y* = −0.00494986 + 0.0483704*x*	0.998	1.22	0.73	1–50	237.5/265.9	0.9969	1.0040	0.0184	3
6	Protocatechuic acid	6.8	152.8	108.0	Negative	*y* = 0.211373 + 12.8622*x*	0.957	1.43	0.76	0.1–5	21.9/38.6	0.9972	1.0055	0.0350	1
7	Catechin	7.4	288.8	203.1	Negative	*y* = −0.00370053 + 0.431369*x*	0.999	2.14	1.08	0.2–10	55.0/78.0	1.0024	1.0045	0.0221	3
8	Gentisic acid	8.3	152.8	109.0	Negative	*y* = −0.0238983 + 12.1494*x*	0.997	1.81	1.22	0.1–5	18.5/28.2	0.9963	1.0077	0.0167	1
9	Chlorogenic acid	8.4	353.0	85.0	Negative	*y* = 0.289983 + 36.3926*x*	0.995	2.15	1.52	0.1–5	13.1/17.6	1.0000	1.0023	0.0213	1
10	Protocatechuic aldehyde	8.5	137.2	92.0	Negative	*y* = 0.257085 + 25.4657*x*	0.996	2.08	0.57	0.1–5	15.4/22.2	1.0002	0.9988	0.0396	1
11	Tannic acid	9.2	182.8	78.0	Negative	*y* = 0.0126307 + 26.9263*x*	0.999	2.40	1.16	0.05–2.5	15.3/22.7	0.9970	0.9950	0.0190	1
12	Epigallocatechin gallate	9.4	457.0	305.1	Negative	*y* = −0.0380744 + 1.61233*x*	0.999	1.30	0.63	0.2–10	61.0/86.0	0.9981	1.0079	0.0147	3
13	1,5-Dicaffeoylquinic acid	9.8	515.0	191.0	Negative	*y* = −0.0164044 + 16.6535*x*	0.999	2.42	1.48	0.1–5	5.8/9.4	0.9983	0.9997	0.0306	1
14	4-OH Benzoic acid	10.5	137.2	65.0	Negative	*y* = −0.0240747 + 5.06492*x*	0.999	1.24	0.97	0.2–10	68.4/88.1	1.0032	1.0068	0.0237	1
15	Epicatechin	11.6	289.0	203.0	Negative	*y* = −0.0172078 + 0.0833424*x*	0.996	1.47	0.62	1–50	139.6/161.6	1.0013	1.0012	0.0221	3
16	Vanillic acid	11.8	166.8	108.0	Negative	*y* = −0.0480183 + 0.779564*x*	0.999	1.92	0.76	1–50	141.9/164.9	1.0022	0.9998	0.0145	1
17	Caffeic acid	12.1	179.0	134.0	Negative	*y* = 0.120319 + 95.4610*x*	0.999	1.11	1.25	0.05–2.5	7.7/9.5	1.0015	1.0042	0.0152	1
18	Syringic acid	12.6	196.8	166.9	Negative	*y* = −0.0458599 + 0.663948*x*	0.998	1.18	1.09	1–50	82.3/104.5	1.0006	1.0072	0.0129	1
19	Vanillin	13.9	153.1	125.0	Positive	*y* = 0.00185898 + 20.7382*x*	0.996	1.10	0.85	0.1–5	24.5/30.4	1.0009	0.9967	0.0122	1
20	Syringic aldehyde	14.6	181.0	151.1	Negative	*y* = −0.0128684 + 7.90153*x*	0.999	2.51	0.77	0.4–20	19.7/28.0	1.0001	0.9964	0.0215	1
21	Daidzin	15.2	417.1	199.0	Positive	*y* = 9.45747 + 152.338*x*	0.996	2.25	1.32	0.05–2.5	7.0/9.5	0.9955	1.0017	0.0202	2
22	Epicatechin gallate	15.5	441.0	289.0	Negative	*y* = −0.0142216 + 1.06768*x*	0.997	1.63	1.28	0.1–5	19.5/28.5	0.9984	0.9946	0.0229	3
23	Piceid	17.2	391.0	135.0/106.9	Positive	*y* = 0.00772525 + 25.4181*x*	0.999	1.94	1.16	0.05–2.5	13.8/17.8	1.0042	0.9979	0.0199	1
24	*p*-Coumaric acid	17.8	163.0	93.0	Negative	*y* = 0.0249034 + 18.5180*x*	0.999	1.92	1.43	0.1–5	25.9/34.9	1.0049	1.0001	0.0194	1
25	Ferulic acid-D3-IS *^h^*	18.8	196.2	152.1	Negative	N.A.	N.A.	N.A.	N.A.	N.A.	N.A.	N.A.	N.A.	0.0170	1
26	Ferulic acid	18.8	192.8	149.0	Negative	*y* = −0.0735254 + 1.34476*x*	0.999	1.44	0.53	1–50	11.8/15.6	0.9951	0.9976	0.0181	1
27	Sinapic acid	18.9	222.8	193.0	Negative	*y* = −0.0929932 + 0.836324*x*	0.999	1.45	0.52	0.2–10	65.2/82.3	1.0031	1.0037	0.0317	1
28	Coumarin	20.9	146.9	103.1	Positive	*y* = 0.0633397 + 136.508*x*	0.999	2.11	1.54	0.05–2.5	214.2/247.3	0.9950	0.9958	0.0383	1
29	Salicylic acid	21.8	137.2	65.0	Negative	*y* = 0.239287 + 153.659*x*	0.999	1.48	1.18	0.05–2.5	6.0/8.3	0.9950	0.9998	0.0158	1
30	Cynaroside	23.7	447.0	284.0	Negative	*y* = 0.280246 + 6.13360*x*	0.997	1.56	1.12	0.05–2.5	12.1/16.0	1.0072	1.0002	0.0366	2
31	Miquelianin	24.1	477.0	150.9	Negative	*y* = −0.00991585 + 5.50334*x*	0.999	1.31	0.95	0.1–5	10.6/14.7	0.9934	0.9965	0.0220	2
33	Rutin	25.6	608.9	301.0	Negative	*y* = −0.0771907 + 2.89868*x*	0.999	1.38	1.09	0.1–5	15.7/22.7	0.9977	1.0033	0.0247	2
34	Isoquercitrin	25.6	463.0	271.0	Negative	*y* = −0.111120 + 4.10546*x*	0.998	2.13	0.78	0.1–5	8.7/13.5	1.0057	0.9963	0.0220	2
35	Hesperidin	25.8	611.2	449.0	Positive	*y* = 0.139055 + 13.2785*x*	0.999	1.84	1.35	0.1–5	19.0/26.0	0.9967	1.0043	0.0335	2
36	*o*-Coumaric acid	26.1	162.8	93.0	Negative	*y* = 0.00837193 + 11.2147*x*	0.999	2.11	1.46	0.1–5	31.8/40.4	1.0044	0.9986	0.0147	1
37	Genistin	26.3	431.0	239.0	Negative	*y* = 1.65808 + 7.57459*x*	0.991	2.01	1.28	0.1–5	14.9/21.7	1.0062	1.0047	0.0083	2
38	Rosmarinic acid	26.6	359.0	197.0	Negative	*y* = −0.0117238 + 8.04377*x*	0.999	1.24	0.86	0.1–5	16.2/21.2	1.0056	1.0002	0.0130	1
39	Ellagic acid	27.6	301.0	284.0	Negative	*y* = 0.00877034 + 0.663741*x*	0.999	1.57	1.23	0.4–20	56.9/71.0	1.0005	1.0048	0.0364	1
40	Cosmosiin	28.2	431.0	269.0	Negative	*y* = −0.708662 + 8.62498*x*	0.998	1.65	1.30	0.1–5	6.3/9.2	0.9940	0.9973	0.0083	2
41	Quercitrin	29.8	447.0	301.0	Negative	*y* = −0.00153274 + 3.20368*x*	0.999	2.24	1.16	0.1–5	4.8/6.4	0.9960	0.9978	0.0268	2
42	Astragalin	30.4	447.0	255.0	Negative	*y* = 0.00825333 + 3.51189*x*	0.999	2.08	1.72	0.1–5	6.6/8.2	0.9968	0.9957	0.0114	2
43	Nicotiflorin	30.6	592.9	255.0/284.0	Negative	*y* = 0.00499333 + 2.62351*x*	0.999	1.48	1.23	0.05–2.5	11.9/16.7	0.9954	1.0044	0.0108	2
44	Fisetin	30.6	285.0	163.0	Negative	*y* = 0.0365705 + 8.09472*x*	0.999	1.75	1.19	0.1–5	10.1/12.7	0.9980	1.0042	0.0231	3
45	Daidzein	34.0	253.0	223.0	Negative	*y* = −0.0329252 + 6.23004*x*	0.999	2.18	1.73	0.1–5	9.8/11.6	0.9926	0.9963	0.0370	3
47	Quercetin	35.7	301.0	272.9	Negative	*y* = +0.00597342 + 3.39417*x*	0.999	1.89	1.38	0.1–5	15.5/19.0	0.9967	0.9971	0.0175	3
48	Naringenin	35.9	270.9	119.0	Negative	*y* = −0.00393403 + 14.6424*x*	0.999	2.34	1.69	0.1–5	2.6/3.9	1.0062	1.0020	0.0392	3
49	Hesperetin	36.7	301.0	136.0/286.0	Negative	*y* = +0.0442350 + 6.07160*x*	0.999	2.47	2.13	0.1–5	7.1/9.1	0.9998	0.9963	0.0321	3
50	Luteolin	36.7	284.8	151.0/175.0	Negative	*y* = −0.0541723 + 30.7422*x*	0.999	1.67	1.28	0.05–2.5	2.6/4.1	0.9952	1.0029	0.0313	3
51	Genistein	36.9	269.0	135.0	Negative	*y* = −0.00507501 + 12.1933*x*	0.999	1.48	1.19	0.05–2.5	3.7/5.3	1.0069	1.0012	0.0337	3
52	Kaempferol	37.9	285.0	239.0	Negative	*y* = −0.00459557 + 3.13754*x*	0.999	1.49	1.26	0.05–2.5	10.2/15.4	0.9992	0.9990	0.0212	3
53	Apigenin	38.2	268.8	151.0/149.0	Negative	*y* = 0.119018 + 34.8730*x*	0.998	1.17	0.96	0.05–2.5	1.3/2.0	0.9985	1.0003	0.0178	3
54	Amentoflavone	39.7	537.0	417.0	Negative	*y* = 0.727280 + 33.3658*x*	0.992	1.35	1.12	0.05–2.5	2.8/5.1	0.9991	1.0044	0.0340	3
55	Chrysin	40.5	252.8	145.0/119.0	Negative	*y* = −0.0777300 + 18.8873*x*	0.999	1.46	1.21	0.05–2.5	1.5/2.8	0.9922	1.0050	0.0323	3
56	Acacetin	40.7	283.0	239.0	Negative	*y* = −0.559818 + 163.062*x*	0.997	1.67	1.28	0.02–1	1.5/2.5	0.9949	1.0011	0.0363	3

*^a^* R.T.: retention time, *^b^* MI (*m*/*z):* molecular ions of the standard analytes (m/z ratio), *^c^* FI (*m*/*z):* fragment ions *^d^ r*^2^: coefficient of determination, *^e^ RSD*: relative standard deviation, *^f^ LOD*/*LOQ* (µg/L): limit of detection/quantification, *^g^ U* (%): relative uncertainty percentage at the 95% confidence level (*k* = 2), *^h^* IS: internal standard, *^i^* Gr. No: represents the grouping of internal standards; these numbers indicate which IS stands for which phenolic compounds.

**Table 2 molecules-28-01739-t002:** LC–MS/MS parameters of selected compounds and amount of antioxidants in propolis in mg/g concentration.

Compounds	Analyte	Phenolics (mg Analyte/g Propolis)
**1**	Quinic acid	7.285
**2**	Fumaric aid	0.134
**3**	Aconitic acid	N.D.
**4**	Gallic acid	0.150
**5**	Epigallocatechin	N.D.
**6**	Protocatechuic acid	1,158
**7**	Catechin	N.D.
**8**	Gentisic acid	N.D.
**9**	Chlorogenic acid	0.018
**10**	Protocatechuic aldehyde	0.338
**11**	Tannic acid	N.D.
**12**	Epigallocatechin gallate	N.D.
**13**	1,5-Dicaffeoylquinic acid	N.D.
**14**	4-OH Benzoic acid	0.306
**15**	Epicatechin	N.D.
**16**	Vanillic acid	1.647
**17**	Caffeic acid	21.358
**18**	Syringic acid	N.D.
**19**	Vanillin	0.116
**20**	Syringic aldehyde	N.D.
**21**	Daidzin	N.D.
**22**	Epicatechin gallate	N.D.
**23**	Piceid	N.D.
**24**	p-Coumaric acid	16.911
**25**	Ferulic acid-D3-IS	N.D.
**26**	Ferulic acid	5.110
**27**	Sinapic acid	N.D.
**28**	Coumarin	N.D.
**29**	Salicylic acid	N.D.
**30**	Cyranoside	N.D.
**31**	Miquelianin	0.020
**32**	Rutin-D3-IS	N.D.
**33**	Rutin	0.035
**34**	Isoquercitrin	0.045
**35**	Hesperidin	0.045
**36**	O-Coumaric acid	N.D.
**37**	Genistin	N.D.
**38**	Rosmarinic acid	N.D.
**39**	Ellagic acid	N.D.
**40**	Cosmosiin	0.026
**41**	Quercitrin	0.087
**42**	Astragalin	0.030
**43**	Nicotiflorin	0.024
**44**	Fisetin	N.D.
**45**	Daidzein	N.D.
**46**	Quercetin-D3-IS	N.D.
**47**	Quercetin	6.223
**48**	Naringenin	11.340
**49**	Hesperetin	2.089
**50**	Luteolin	4.394
**51**	Genistein	N.D.
**52**	Kaempferol	4.043
**53**	Apigenin	4.686
**54**	Amentoflavone	N.D.
**55**	Chrysin	9.860
**56**	Acacetin	76.359

N.D.: not detected, IS: internal standard.

**Table 3 molecules-28-01739-t003:** The reducing abilities of propolis and standards a concentration of 30 μg/mL.

Antioxidants	Fe^3+^-Reducing		Cu^2+^-Reducing		FRAP-Reducing	
	λ_700_	r^2^	λ_700_	r^2^	λ_700_	r^2^
BHA	1.257 ± 0.088	0.9523	1.800 ± 0.156	0.9742	0.884 ± 0.116	0.9899
BHT	2.018 ± 0.029	0.9466	2.912 ± 0.012	0.9969	2.089 ± 0.027	0.9581
α-Tocopherol	1.895 ± 0.008	0.9402	1.139 ± 0.096	0.9967	1.995 ± 0.016	0.9807
Trolox	1.545 ± 0.019	0.9966	2.323 ± 0.049	0.9980	1.755 ± 0.093	0.9990
Propolis	0.894 ± 0.020	0.9953	0.778 ± 0.054	0.9986	1.114 ± 0.045	0.9970

**Table 4 molecules-28-01739-t004:** The reducing abilities of propolis and standards at concentration of 30 μg/mL.

Antioxidants	DPPH^•^ Scavenging		ABTS^•+^ Scavenging		DMPD^•+^ Scavenging	
	IC_50_	r^2^	IC_50_	r^2^	IC_50_	r^2^
BHA	9.00	0.9399	7.71	0.9330	31.43	0.9993
BHT	21.00	0.9668	7.71	0.9330	-	-
α-Tocopherol	5.92	0.9770	7.71	0.9330	14.38	0.9349
Trolox	9.63	0.9947	8.10	0.9550	-	-
Propolis	20.55	0.9989	8.157	0.9985	86.64	0.9855

**Table 5 molecules-28-01739-t005:** Half-maximal inhibition concentration (IC_50_, μg/mL) of propolis against human carbonic anhydrase isoenzyme II (hCA II) acetylcholinesterase (AChE), and α-glycosidase (α-Gly) enzymes.

Antioxidants	hCA II		AChE		α-Glycosidase	
	IC_50_	r^2^	IC_50_	r^2^	IC_50_	r^2^
Propolis	19.6	0.9327	3.4	0.9869	3.7	0.9362
Acetazolamide *	8.37	0.9825	-	-	-	-
Tacrine **	-	-	5.97	0.9706	-	-
Acarbose ***	-	-	-	-	22,800	-

*Acetazolamide was used as standard inhibitor for hCA, ** Tacrine was used as standard inhibitor for AChE, *** Acarbose was as used standard inhibitor for α-glycosidase, which was obtained from the literature [50].

## Data Availability

Data are provided in a publicly accessible repository.

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
