# Peer review of "Comprehensive Metabolite Profiling of Berdav Propolis Using LC-MS/MS: Determination of Antioxidant, Anticholinergic, Antiglaucoma, and Antidiabetic Effects"

_molecules, 2023, doi:10.3390/molecules28041739_

Round 1

Reviewer 1 Report

The authors study the metabolites of Prorplis. This is one of the common foods in Turkey. Studying components and biological effects will be beneficial for future health and aging research scene. There are some points that the authors should explain and revise in the present form.

Major points

1)           The authors did in silico analyses for acacetin to three enzymes. However, the authors did not assay acacetin with these three enzyme inhibition assays. Generally, in silico analyses are supportive, and the real wet data is more assertive. If the compounds can not inhibit the enzymes really, the computational analyses have no mean.

Acacetin can be purchased from the following suppliers

Angene International Limited: AG00I8QX

Thermo scientificAcros Organics: 465232500

MedChem Express: HY-N0451

If the authors cannot add this data, the authors should remove in silico analyses. The other data is meaningful and sufficient for publication without in silico analyses, I think.

2)           Results should be subdivided by kinds of experiments.

3)           Lines in Figures should be more bold.

4)           Not use curved lines in Figures. Kindly connect all points with straight lines.

5)           Please site Reference appropriately. Please re-check all. For example, the following is the point only the reviewer noticed.

In Lines 60-62, the authors wrote the components of propolis 5% pollen, 5% wax, 30% resins and plant balsams, 10% essential oils, 50 % resins and plant balsams. The remaining 5% is made up of various elements, including organic compounds [7,8].

Why “resins and plant balsams” was described twice? Why total % is 105%? To check this point, the reviewer searched references 7 and 8. First, Ref 8 lacks information about pages. Please check all references. The authors can use computer software that supports adding references, such as Endnote, Mendeley, and Zotero. The latter two software are free to use. By the way, Reference 7 and 8 describe about the components of propolis as

In general, this complex mixture is composed of around 50% resins and plant balsams, 30% wax, 10% essential oils, 5% pollen, and 5% of other substances and materials, including organic compounds.

Please check that point and re-check the whole manuscript, again

6) In materials and methods section, titles of some enzymatic analyses are inappropriate. Antidiabetic Assay -- >α-glucosidase enzyme inhibition assay, Anticholinergic assay -- > AChE enzyme inhibition assay , Antigloucoma assay -- > human CA II enzyme inhibition assay

The subsections are pieced in different manners.

Minor points

1)           L251, Table 4 should be Table 5

2)           Why the order of showing results in Table 5 was different in the plain text? It is difficult to read.

3)           Writing check is needed. For example, the following;

L53: most significant.4 -- > [4]

L386: [90].99 -- > [90-99]??

(descriptions of references should be checked)

L357: 376,73 -- > 376.73

L452: 2,1 -- > 2.1

(descriptions of decimal points should be unified throughout the whole manuscript)

In summary, this manuscript is not suitable for publication in Molecules. However, if the authors rewrite the above-mentioned points, the manuscript will be a valuable report for the world’s readers and be approved for reviewing process in Molecules.

Author Response

RESPONSES TO REVIEWER-1

  • The authors study the metabolites of Prorplis. This is one of the common foods in Turkey. Studying components and biological effects will be beneficial for future health and aging research scene. There are some points that the authors should explain and revise in the present form.

RESPONSE: We would like to thank the referee who liked our article and gave a positive opinion.

Major points

1) The authors did in silico analyses for acacetin to three enzymes. However, the authors did not assay acacetin with these three enzyme inhibition assays. Generally, in silico analyses are supportive, and the real wet data is more assertive. If the compounds cannot inhibit the enzymes really, the computational analyses have no mean.

Acacetin can be purchased from the following suppliers

Angene International Limited: AG00I8QX

Thermo scientific (Acros Organics): 465232500

MedChem Express: HY-N0451

If the authors cannot add this data, the authors should remove in silico analyses. The other data is meaningful and sufficient for publication without in silico analyses, I think.

RESPONSE: The reviewer is right. We removed all in silico analyses from the revised manuscript.

2) Results should be subdivided by kinds of experiments.

RESPONSE: The results section was subdivided to five kinds of experiments including “2.1. Phenolic contents of propolis”, “2.2. Reducing abilities results”, “2.3. Radical scavenging results”, “2.4. Enzyme inhibition results” and “2.5. Molecular docking studies rsults”.

3) Lines in Figures should be more bold.

RESPONSE: Figures 2 and 3 were deleted from text according to academic editor’s recommendation.

4) Not use curved lines in Figures. Kindly connect all points with straight lines.

RESPONSE: Figures 2 and 3 were deleted from text according to academic editor’s recommendation.

5) Please site Reference appropriately. Please re-check all. For example, the following is the point only the reviewer noticed.

RESPONSE: References were checked again and necessary corrections were made.

In Lines 60-62, the authors wrote the components of propolis 5% pollen, 5% wax, 30% resins and plant balsams, 10% essential oils, 50 % resins and plant balsams. The remaining 5% is made up of various elements, including organic compounds [7,8].

Why “resins and plant balsams” was described twice? Why total % is 105%? To check this point, the reviewer searched references 7 and 8. First, Ref 8 lacks information about pages. Please check all references. The authors can use computer software that supports adding references, such as Endnote, Mendeley, and Zotero. The latter two software are free to use. By the way, Reference 7 and 8 describe about the components of propolis as

In general, this complex mixture is composed of around 50% resins and plant balsams, 30% wax, 10% essential oils, 5% pollen, and 5% of other substances and materials, including organic compounds.

RESPONSE: This information was corrected as “Overall, this complex combination is made up of 50% resins, 30% wax, 10% essential oils, 5% pollen, and 5% of other substances and materials, including organic compounds”. In this way, the total content is 100%. For reference 8, the full information of the article is given. We used Endnote, Mendeley, or Zotero before, but we stopped using it because it caused some confusion. We will learn to use the programs fully.

Please check that point and re-check the whole manuscript, again

RESPONSE: We checked the specified point together with the entire draft and made the necessary corrections.

6) In materials and methods section, titles of some enzymatic analyses are inappropriate. Antidiabetic Assay -- >α-glucosidase enzyme inhibition assay, Anticholinergic assay -- > AChE enzyme inhibition assay, Antigloucoma assay -- > human CA II enzyme inhibition assay.

The subsections are pieced in different manners.

RESPONSE: The titles of “Antidiabetic Assay” was changed as “α-glycosidase enzyme inhibition assay”. “Anticholinergic assay” was changed as “AChE enzyme inhibition assay” and “Antigloucoma assay” was changed as “hCA II isoenzyme inhibition assay”.

The subsections were tried to be given in the same way.

Minor points

1) L251, Table 4 should be Table 5

RESPONSE: “Table 4” was corrected as “Table 5”.

2) Why the order of showing results in Table 5 was different in the plain text? It is difficult to read.

RESPONSE: The required corrections were made. The order of the enzymes in Table 5 and in the text was also rearranged.

3) Writing check is needed. For example, the following;

L53: most significant.4 -- > [4]

RESPONSE: “4” was given there by mistake, it was deleted from the text.

L386: [90].99 -- > [90-99]??

RESPONSE: “99” was given there by mistake, it was deleted from the text as “[90].”

(descriptions of references should be checked)

RESPONSE: The descriptions of references was checked again.

L357: 376,73 -- > 376.73

RESPONSE: “376,73” was corrected as “376.73”.

L452: 2,1 -- > 2.1

RESPONSE: “2,1” was corrected as “2.1”.

(descriptions of decimal points should be unified throughout the whole manuscript)

RESPONSE: The descriptions of decimal points were unified throughout the whole manuscript.

In summary, this manuscript is not suitable for publication in Molecules. However, if the authors rewrite the above-mentioned points, the manuscript will be a valuable report for the world’s readers and be approved for reviewing process in Molecules.

RESPONSE: Many thanks to the reviewer due to his/her valuable recommendations and contributions. All of suggested corrections were made point by point. We also believe that after these corrections, our article will be a good reference for the valuable report for the world's readers, especially "Molecules" readers.

Reviewer 2 Report

It gives me pleasure to accept this invitation to review this manuscript. The manuscript entitled (Comprehensive Metabolite Profiling of Berdav Propolis Using LC-MS/MS: Determination of Antioxidant, Anticholinergic, Antiglaucoma, and Antidiabetic Effects).The authors do valuable work, but there are some mistakes I determined in the attached manuscript.

- The introduction is too long. It is better to be somewhat short.

- The sample details should be written: time and date of collection. How much did you collect? Berdav proplis in the title of the manuscript should be mentioned in section of extract preparation

- For preparation of the PEE: The yield should be calculated.

- I need explain in page 9:  in extract preparation Both (What do the authors mean?) only PEE was prepared. Do you mean the samples prepared in DMSO. Please, rewrite again this part. and write the obtained residue weight.

Author Response

RESPONSES TO REVIEWER-2

  • It gives me pleasure to accept this invitation to review this manuscript. The manuscript entitled (Comprehensive Metabolite Profiling of Berdav Propolis Using LC-MS/MS: Determination of Antioxidant, Anticholinergic, Antiglaucoma, and Antidiabetic Effects). The authors do valuable work, but there are some mistakes I determined in the attached manuscript.

RESPONSE: First of all, we would like to thank the referee for her positive comments about us and our article. We also tried to make the desired corrections point by point.

- The introduction is too long. It is better to be somewhat short.

RESPONSE: The introduction was considerably shortened. For this purpose, the following information was deleted from the revised manuscript:

            “Bees utilize propolis as a protective barrier against harmful germs and as a sealing wax to fill up beehive fractures. As a "chemical weapon," it is regarded as being the most significant. Geographical locations, botanical sources, and bee species all have a significant impact on the chemical makeup of propolis [3,4].”.

            “Propolis’s rich and diverse chemical makeup has led to the identification of a large number of substances, including triterpenes, aldehydes, lignans, chalcones, phenolic acids, flavonoids, and sugars. But the most common types are phenolics [9]. Since the beginning of human existence, propolis has been utilized in traditional medicine and has become popular among many different cultures, including the Egyptians, Arabs, and Greeks [10].”.

            “The microbiological durability and quality of foods during storage of meat, fruits, vegetables, milk, etc. have been claimed to be improved by propolis extracts, particularly ethanol and water extracts. The used extracts also contribute to the chemical and physical characteristics of food. Additionally, propolis extracts are used to food packaging [13].”

            “Propolis is a desirable natural product to use as a functional ingredient in meals despite the variations in the antioxidant and pharmacological activities found in propolis from across the world [30]. Considering the plant sources readily accessible to bees for their production, propolis is said to be an excellent example of a naturally occurring combination of antioxidant chemicals, whose content is very diverse and depending on the origin of the sample [31]. One of the top producers of honey worldwide is Turkey. Researchers focused on propolis samples taken from various parts of Turkey because to its extensive production and range of products, as well as its significant chemical and biological qualities [32].

- The sample details should be written: time and date of collection. How much did you collect? Berdav propolis in the title of the manuscript should be mentioned in section of extract preparation

RESPONSE: For this aim, the following information was given in the indicated section: “A sample of propolis (50 g) was collected in August 2022 from one of the beehives of Yuksel Gulcin, a farmer, located in Berdav village of Tutak district of Agri and stored there before processing.”.

- For preparation of the PEE: The yield should be calculated.

RESPONSE: The yield of propolis extraction was calculated by using Eq. (1): “Yield = weight of propolis extract (g) /weight of raw propolis (g) ×100%”, “Yield= 18.25/75*100=75%”. The yield of propolis was 75% and 18.75 g extract was’’ added to extract preparation section.

- I need explain in page 9:  in extract preparation Both (What do the authors mean?) only PEE was prepared. Do you mean the samples prepared in DMSO. Please, rewrite again this part. and write the obtained residue weight.

RESPONSE: This statement was also a mistake; it has been corrected in the text. This part has been rearranged.

Reviewer 3 Report

Dear Reviewer,

The manuscript “Comprehensive Metabolite Profiling of Berdav Propolis Using 2 LC-MS/MS: Determination of Antioxidant, Anticholinergic, Antiglaucoma, and Antidiabetic Effects” is very well written and is good contribution. I recommend this manuscript for publication after following minor revisions

Abstract

The IC50 values of propolis in the ABTS + , DPPH• and DMPD•+ scavenging activity are as follows; remove follow and write it as ‘DMPD•+ scavenging activity are as;’

Introduction

terpenoid, and alkaloids components ‘Remove comma after terpenoids’ Comma and and can’t be used together

antiseptic, and antioxidant characteristics ‘same issue with this sentences please check this and make correction throughout manuscript’

Results

Figure 4 is not clear. It is suggested to insert clearer diagram.

p-coumaric acid, and caffeic acid, ferulic acid, phenylethyl ester are among the 316 ingredients of propolis “remove and from sentence and add it before phenylethyl ester”

For enzyme glycosidase, propolis has IC50 values 3.7 μg/mL “Rewrite the sentence”

Rewrite paragraph ‘In addition, CA II is frequently linked to a number of illnesses’

The figures for the values of Anticholinergic, antiglaucoma, and antidiabetic effects are missing please provide the figures along with the reference used for the studies.

Author Response

RESPONSES TO REVIEWER-3

  • The manuscript “Comprehensive Metabolite Profiling of Berdav Propolis Using 2 LC-MS/MS: Determination of Antioxidant, Anticholinergic, Antiglaucoma, and Antidiabetic Effects” is very well written and is good contribution. I recommend this manuscript for publication after following minor revisions

RESPONSE: We would like to thank our referee for liking our manuscript and giving a positive opinion.

Abstract

The IC50 values of propolis in the ABTS‧ + , DPPH• and DMPD•+ scavenging activity are as follows; remove follow and write it as ‘DMPD•+ scavenging activity are as;’

RESPONSE: this sentence was corrected as “The IC50 values of propolis in the ABTS‧+, DPPH‧ and DMPD‧+ scavenging activity was found 8.157, 20.55 and 86.64 μg/mL, respectively.”

Introduction

terpenoid, and alkaloids components ‘Remove comma after terpenoids’ Comma and and can’t be used together

RESPONSE: It was corrected.

antiseptic, and antioxidant characteristics ‘same issue with this sentences please check this and make correction throughout manuscript’

RESPONSE: It was corrected.

Results

Figure 4 is not clear. It is suggested to insert clearer diagram.

RESPONSE: We removed all in silico analyses from the revised manuscript according to the recommendation of Reviewer 1. So, Figure 4 was deleted from the text.

p-coumaric acid, and caffeic acid, ferulic acid, phenylethyl ester are among the 316 ingredients of propolis “remove and from sentence and add it before phenylethyl ester”

RESPONSE: This sentence is not fully understood. This sentence was rearranged as “Caffeic acid phenylethyl ester, gallic acid, cinnamic acid, galangin, caffeic acid, naringenin, luteolin, kaempferol, quercetin, pinocembrin, rutin, p-coumaric acid, and, ferulic acid, are among the ingredients of propolis. They ultimately enhance…”.

For enzyme glycosidase, propolis has IC50 values 3.7 μg/mL “Rewrite the sentence”

RESPONSE: The sentence of “For enzyme glycosidase, Propolis has IC50 values 3.7 μg/mL (Table 5)” was corrected as “Propolis extracts displayed IC50 value of 3.7 μg/mL towards α-glycosidase enzyme (r2: 0.9362, Table 5)”.

Rewrite paragraph ‘In addition, CA II is frequently linked to a number of illnesses’

RESPONSE: The sentence of “In addition, CA II is frequently linked to a number of illnesses, including glaucoma, osteoporosis, and renal tubular acidosis.” was corrected as “The hCA II isoform is associated with some disorders including glaucoma, osteoporosis, and renal tubular acidosis.”

The figures for the values of Anticholinergic, antiglaucoma, and antidiabetic effects are missing please provide the figures along with the reference used for the studies.

RESPONSE: “Figure 2. The half maximal inhibitory concentration (IC50) graphs of propolis against human carbonic anhydrase II (hCA II) (A), acetylcholinesterase (AChE) (B) and α-glycosidase (C) enzymes.” was provided for this aim.

Round 2

Reviewer 1 Report

The reviewer is glad to review this manuscript again. The authors have corrected the article properly for the most part. The manuscript needs to revise kindly on the following point.

Major points

1.      Subsection 4.13 and 4.14 might be reversed. The content of 4.13 has indicated AChE enzyme inhibition assay, and 4.14 has been α-Glycosidase enzyme inhibition assay.

2.      The authors discuss the general properties of AChE enzyme inhibition, α-Glycosidase, and hCAII in the Discussion section. However, other research about propolis against each enzyme inhibition should be referred to in the Discussion section (for example, below). The authors have to introduce to the Molecules readers at least these previous reports by the other researchers, and clarify own position in the world. Moreover, if possible, the authors should write novelty points of this Turkish propolis (or this research) compared to the other reported propolis in other production areas to Discussion or Conclusion.

(For example)

Milena Popova, et al.

Nat Prod Commun. 2015 Nov;10(11):1961-4.

Antioxidant and α-Glucosidase Inhibitory Properties and Chemical Profiles of Moroccan Propolis

https://pubmed.ncbi.nlm.nih.gov/26749837/

Nimet Baltas et al.,

J Enzyme Inhib Med Chem. 2016;31(sup1):52-55. doi: 10.3109/14756366.2016.1167049. Epub 2016 Apr 7. Inhibition properties of propolis extracts to some clinically important enzymes

https://pubmed.ncbi.nlm.nih.gov/27052345/

Imdat Aygul

J Enzyme Inhib Med Chem. 2016;31(sup4):119-124. doi: 10.1080/14756366.2016.1221406. Epub 2016 Aug 25.Investigation of the inhibitory properties of some phenolic standards and bee products against human carbonic anhydrase I and II

https://pubmed.ncbi.nlm.nih.gov/27559016/

Minor points: In 4.17, Line 516, “foe each sample” may be “for each sample”

Author Response

RESPONSES TO REVIEWER-1

  • The reviewer is glad to review this manuscript again. The authors have corrected the article properly for the most part. The manuscript needs to revise kindly on the following point.

RESPONSE: We would like to thank the reviewer due to his/her positive opinion.

Major points

  1. 1. Subsection 4.13 and 4.14 might be reversed. The content of 4.13 has indicated AChE enzyme inhibition assay, and 4.14 has been α-Glycosidase enzyme inhibition assay.

RESPONSE: Thank you very much for the referee's attention. Both titles have been corrected.

  1. 2. The authors discuss the general properties of AChE enzyme inhibition, α-Glycosidase, and hCA II in the Discussion section. However, other research about propolis against each enzyme inhibition should be referred to in the Discussion section (for example, below). The authors have to introduce to the Molecules readers at least these previous reports by the other researchers, and clarify own position in the world. Moreover, if possible, the authors should write novelty points of this Turkish propolis (or this research) compared to the other reported propolis in other production areas to Discussion or Conclusion.

(For example)

Milena Popova, et al.

Nat Prod Commun. 2015 Nov;10(11):1961-4.

Antioxidant and α-Glucosidase Inhibitory Properties and Chemical Profiles of Moroccan Propolis

https://pubmed.ncbi.nlm.nih.gov/26749837/

Nimet Baltas et al.,

J Enzyme Inhib Med Chem. 2016;31(sup1):52-55. doi: 10.3109/14756366.2016.1167049.

Inhibition properties of propolis extracts to some clinically important enzymes

https://pubmed.ncbi.nlm.nih.gov/27052345/

Imdat Aygul

J Enzyme Inhib Med Chem. 2016;31(sup4):119-124. doi: 10.1080/14756366.2016.1221406. Investigation of the inhibitory properties of some phenolic standards and bee products against human carbonic anhydrase I and II

https://pubmed.ncbi.nlm.nih.gov/27559016/

RESPONSE: Discussion section was re-arranged. Our study was compared to the other reported Turkish propolis in other production areas in Discussion section. Also the following references were argued and given in our revised manuscript:

Popova, M.; Lyoussi, B.; Aazza, S.; Antunes, D.; Bankova, V.; Miguel, G. Antioxidant and α-glucosidase inhibitory properties and chemical profiles of Moroccan propolis. Nat. Prod. Commun. 2015, 10(11), 1961-1964.

Baltas, N.; Yildiz, O.; Kolayli, S. Inhibition properties of propolis extracts to some clinically important enzymes. J. Enzyme Inhib. Med. Chem. 2016, 31(S1), 52-55.

Aygul, I.; Yaylaci Karahalil, K.; Supuran, C.T. Investigation of the inhibitory properties of some phenolic standards and bee products against human carbonic anhydrase I and II. J. Enzyme Inhib. Med. Chem. 2016;31 (S4), 119-124.

Minor points: In 4.17, Line 516, “foe each sample” may be “for each sample”.

RESPONSE: “foe each sample” was corrected as “for each sample”.